# Comparative analysis of viruses in four bee species collected from agricultural, urban, and natural landscapes

**Tugce Olgun**[1], **Sydney E. Everhart**[2], **Troy Anderson**[1], **Judy Wu-Smart**[1]*

**1** Department of Entomology, University of Nebraska, Lincoln, Nebraska, United States of America,
**2** Department of Plant Pathology, University of Nebraska, Lincoln, Nebraska, United States of America

* jwu-smart@unl.edu

**Data Availability Statement:** All data are available in the UNL Data Repository system. Karacoban, T and Wu-Smart, J. (2019). "Comparative analysis of viruses in four bee species collected from

## Abstract

Managed honey bees (*Apis mellifera* L.) and wild bees provide critical ecological services that shape and sustain natural, agricultural, and urban landscapes. In recent years, declines in bee populations have highlighted the importance of the pollination services they provide and the need for more research into the reasons for global bee losses. Several stressors cause declining populations of managed and wild bee species such as habitat degradation, pesticide exposure, and pathogens. Viruses, which have been implicated as a key stressor, are able to infect a wide range of species and can be transmitted both intra- and inter-specifically from infected bee species to uninfected bee species via vertical (from parent to offspring) and/or horizontal (between individuals via direct or indirect contact) transmission. To explore how viruses spread both intra- and inter-specifically within a community, we examined the impact of management, landscape type, and bee species on the transmission of four common viruses in Nebraska: Deformed wing virus (DWV), Israeli acute paralysis virus (IAPV), Black queen cell virus (BQCV), and Sacbrood virus (SBV). Results indicated the prevalence of viruses is significantly affected ($P < 0.005$) by bee species, virus type, and season, but not by landscape or year ($P = 0.290$ and $0.065$ respectively). The higher prevalence of DWV detected across bee species (10.4% on *Apis mellifera*, 5.3% on *Bombus impatiens*, 6.1% on *Bombus griseocollis*, and 22.44% on *Halictus ligatus*) and seasons (10.8% in early-mid summer and 11.4% in late summer) may indicate a higher risk of interspecific transmission of DWV. However, IAPV was predominately detected in *Halictus ligatus* (20.7%) and in late season collections (28.1%), which may suggest species-specific susceptibility and seasonal trends in infection rates associated with different virus types. However, there were limited detections of SBV and BQCV in bees collected during both sampling periods, indicating SBV and BQCV may be less prevalent among bee communities in this area.

## Introduction

There are roughly 20,000 extant species of bees worldwide and approximately 3,600 species of bees in North American that aid in the pollination of our agricultural crops and native flora

agricultural, urban, and natural landscapes." UNL Data Repository. Dataset. doi: 10.32873/unl.dr. 20191217.

**Funding:** The author(s) received no specific funding for this work.

**Competing interests:** The authors have declared that no competing interests exist.

[1,2]. Domesticated honey bees (*Apis mellifera* L.) play vital roles as the principal providers of food crop pollination, contributing over $15 billion USD in added value [1]. In addition to the services provided by commercially managed honey bees, other species of social and solitary bees are critical for the pollination of ecologically important plants in natural, agricultural, and urban landscapes [1,2]. Many of these wild, unmanaged bees, however, are increasingly being utilized commercially as alternative pollinators for certain crops and/or under more specific environmental conditions. For example, bumble bees (*Bombus impatiens* Cresson and *B. terrestris* L.*)*, are often used for the pollination of greenhouse crops, such as tomatoes (*Lycopersicon esculentum* Mill. [3]), and sweet pepper (*Capsicum annuum* L. var. *grossum* cv. Superset [4]), because bumble bee colonies are smaller and annual, and the bees themselves adapt to indoor settings better than honey bees. Bumble bees are also managed commercially to pollinate cooler climate crops, such as blueberries (*Vaccinium corymbosum* L. [5]), cranberries (*Vaccinium macrocarpon* Ait [6]), due to their ability to forage under lower temperatures. Bumble bees pollinate some plants more efficiently through "buzz" pollination, which is the ability to sonicate flowers by vibrating thoracic muscles while grasping onto flowers. Sonication causes pollen grains secured in plant anthers to dehisce and attach to bee hairs, which more effectively collects pollen than active pollen collection by honey bees [7]. In fact, *Bombus griseocollis* and *B. impatiens* are reportedly able to carry the largest amount of pollen compared to 41 other species of native bees [8]. Furthermore, Halictids, or sweat bees, are another primitively eusocial species like bumble bees and are also efficient pollinators because they exhibit similar foraging traits and appear to be better adapted to agricultural settings [9]. Halictidae is the second largest family of Apoidea bees and many Halictids are commonly reported in agricultural settings even establishing ground nests between crop rows [i.e., *Lasioglossom (Dialictus)*, nesting at high rates between rows of coffee plants in Costa Rica; 10].

Recent studies indicate declining populations of managed and wild bee species over the past century [11,12]. Common factors identified as contributors to both managed and wild bee declines include habitat degradation, pesticide exposure, and pathogens [12,13]. Agricultural intensification and urban growth have dramatically reduced the amount of natural landscapes and diminished floral abundance and diversity, which are critical for the long-term establishment of healthy pollinator communities. With increased crop production and urbanization comes increased risk of pesticide exposure on bees, when growers and homeowners treat for unwanted pests, pathogens, and weeds. Pesticide exposure can have lethal consequences or cause various adverse sub-lethal effects on bees including impaired cognitive behaviors, disrupted foraging activity, suppression of the immune system, and increased susceptibility to other pests and diseases [14]. While there exists extensive literature on pests and pathogens that affect honey bees, there are knowledge gaps in our understanding of pathogen transmission within honey bee colonies and across other species of bees.

Viruses have a broad host range and several viruses can exist in their host in a latent form or be asymptomatic, making monitoring efforts difficult. More than 30 distinct viruses have been identified in honey bees thus far, but this is an evolving area of science that continues to discover new strains [2]. It is well-documented that diseases commonly detected in honey bees can be transmitted to individual nestmates and within colonies, as well as across different bee species and other insects [15]. The occurrence of vertical transmission from adults to offspring and or horizontal transmission between nestmates via direct contact with infected individuals or indirect contact of contaminated foods allows viruses to persist in colonies. Horizontal transmission may occur among nestmates of honey bees interacting with contaminated foods (intraspecific transmission) or/and may occur by physical contact between different bee species during foraging at common site (interspecific transmission) [16,17]. Long-term

persistence of viruses increases the risk of pathogen spread from infected to uninfected individuals or colonies.

The most commonly detected virus in honey bees, Deformed wing virus (DWV), is vectored by a major ectoparasitic mite (*Varroa destructor*) of honey bees, however, DWV is found in both managed bumble bees and other wild bees (*Osmia* spp., *Augochlora* spp., and *Xylocopa* spp.), despite the absence of Varroa mites [18]. Bumble bees and wasps are common robbers of weak honey bee colonies and may serve as suitable hosts for DWV as they come into direct contact with infected honey bees or indirectly through contaminated foods [19,20]. While many bee viruses can only replicate in suitable hosts, some bee viruses, such as Sacbrood virus (SBV) and Black queen cell virus (BQCV), are also capable of replicating in mite vectors (*Varroa* spp. and *Tropilaelaps* spp. [21]).

Nebraska is an important agricultural state particularly for the production of corn and soybeans. Crop coverage in Nebraska has increased from 3.4 to 3.8 million hectares in the last 10 years [22]. The growth of agricultural production in Nebraska and across the nation has led to decreasing availability of habitat and increasing agrochemical exposure for pollinators (e.g., birds, bees, and butterflies) [23–26]. Studies have shown that crop field margins may be utilized to provide important habitat and promote increased insect diversity (including natural predators of crop pests) and food sources for wildlife [27]. Some crops (e.g., alfalfa, vetch, sweet clover, sunflower) grown in Nebraska depend on bees for pollination but this is minor in comparison to the number of wind-pollinated crops (e.g., corn, soybeans, sorghum, wheat) grown in the state. While 97.2% of Nebraska is privately-owned, there are several biologically unique landscapes protected to conserve species, identified by the Nebraska Game and Parks Service as "at-risk" or "threatened", which includes several insects and bees [28]. Therefore, research on drivers of bee loss are critically important for identifying potential bee and land management practices that may promote bee communities and mitigate further bee declines. Knowledge of which viruses are present and how they are transmitted both intra- and interspecifically within the pollinator community will help elucidate the impact of management, landscape type, and bee species on the transmission of viruses. Here, we present a descriptive survey of bee viruses (Deformed wing virus (DWV), Israeli acute paralysis virus (IAPV), Black queen cell virus (BQCV), and Sacbrood virus (SBV)) found in managed bee species (*A. mellifera* and *B. impatiens*) and unmanaged bee species (*B. griseocollis* and *H. ligatus*) that were collected from different landscapes (agricultural, urban, and natural or open spaces) across the growing season. We hypothesized that managed bees would have more frequent detections of viruses due to greater densities of bees in managed apiaries and the persistence of pathogens in social colonies. Likewise, we hypothesized landscapes with more limited floral resources would lead to more concentrated foraging sites and co-mingling across species, thus driving higher rates of interspecific transmission of viruses. Result showed that viral prevalence was affected by virus type and season, but not by landscape. Significant differences between managed and unmanaged bees were observed only in the urban site with positive detections of DWV and IAPV in 2018. Results of this study will be used to improve our understanding of interspecific transmission of viruses from honey bees, which may inform beekeeping and landscape management recommendations to mitigate direct contact of infected bees with wild populations to reduce pathogen spread.

## Materials & methods

### Target bee species and site selection

Four generalist species of bees were selected for this study because of the relative ease of finding them in all landscapes across the season. Targeted bee species included, honey bees [*Apis*

*mellifera* (Apidae)], bumble bees [*Bombus impatiens* and *B. griseocollis* (Apidae)], and sweat bees [*Halictus ligatus* (Halictidae)]. Collections of bees occurred at 10 sites located in agricultural (three sites adjacent to row crops; UNL Eastern Nebraska Research and Extension Center, Goat Farm, and Dixi Farm), urban (four public garden sites; Rose Garden, Sunken Garden, Lauritzen Garden, and UNL Pollinator Garden), and natural (two open space sites; Lincoln Prairie Corridor and Union City roadsides) landscapes, during the summer from May 1$^{st}$ through September 30$^{th}$, over two years (2017 and 2018). Agricultural field sites consisted of two privately-owned farms (Dixi Farm located in Waverly, NE and Goat Farm located in Lincoln, NE) that housed honey bee colonies (owned and managed by University of Nebraska-Lincoln, UNL) and pollinator habitat plots (e.g. cover crops such as clover and alfalfa) established in field margins of research corn and soybean fields at the UNL Eastern Nebraska Research and Extension Center (ENREC) in Mead, NE. Study sites owned by the University of Nebraska-Lincoln (UNL Eastern NE research and Extension Center & UNL Pollinator Garden) did not require access permission. Permission was granted for all other collection sites by the appropriate authorities: Union roadsides (permission granted by Nebraska Department of Transportation), Lauritzen Garden (permission granted by Lauritzen Garden), private lands "Goat Farm" and "Dixi Farm" (permission provided by landowners), and all remaining garden sites are overseen by the City of Lincoln public parks (permission granted from the City of Lincoln). No protected lands were used in this study nor were there formal field permits required. We did not collect or survey any protected species. Foraging honey bees and wild bees were collected at each site either within pollinator habitats or in field margins absent pollinator plantings (S1 Table).

Urban sites included four public gardens within two major cities (Lincoln and Omaha, NE). The UNL Pollinator Garden, the Rose Garden, and the Sunken Garden are all located within the city of Lincoln, while Lauritzen Gardens Botanical Center is located in Omaha, NE. These sites contained pollinator-friendly plantings of both native and cultivated species and ranged in size from <0.40 hectare to >40.5 hectares. Different garden spaces provided varying dietary resources and nesting habitats for wildlife. The carrying capacity for pollinators in gardens was dependent on the plant types, phenology of flowers, and management of gardens. The Rose Garden, for example, was less suitable for pollinators due to the perennial ornamental roses that are cultivated for ornate petals but offer little to no floral nectar or pollen for bees (S1 Table).

The natural area or open space sites included public parks and conservation areas, such as the Lincoln Prairie Corridor, which spans from Pioneers Park (tallgrass prairie ecosystem) to the Spring Creek Audubon Center (shortgrass prairie ecosystem), and public roadsides (Union City, NE). Open spaces also varied in the quality of pollinator habitat they provided. Some sites were newly-seeded (roadside plots) and recently restored (prairie), while other areas remained relatively undisturbed for more than five years and were either plots in the conservation reserve program (CRP) or unmanaged prairie (S1 Table). Additionally, roadside plots had 4–6 commercial *Bombus impatiens* colonies throughout the season along the 11 km stretch of Nebraska Highway 75 for a separate pollinator habitat study. Therefore, we are unable to differentiate from wild and commercially sourced *B. impatiens* and have categorized these bees as "managed" species in this study.

### Foraging bee collection

*Apis mellifera*, *B. impatiens*, *B. griseocollis*, and *H. ligatus* were collected using nets and hand vial trapping. Ten individual foraging bees of each species were collected from each landscape type during summer from May 1$^{st}$ through September 30$^{th}$ in 2017 and 2018. Collections

occurred between 8am–12pm in early, middle, and late summer. In nets, it was possible that physical contact among bees would result in some surface-level cross-contamination, although internal cross-contamination was considered unlikely [29]. To reduce potential for cross-contamination, target bees were separated immediately after collection in individual vials and stored on ice (< 3 hours) in the field until samples could be returned to the laboratory. Bees were stored in a -20°C freezer prior to identification and processing. Identification of bees was confirmed using three identification guides [30–32]. After identification, each bee was placed into a 1.5 ml centrifuged tube and stored at -80°C until processed for virus identification using RT-PCR.

## Honey bee colonies and apiaries

Twenty-six honey bee (*Apis mellifera*) colonies were set up at the UNL Pollinator Garden between May 1st and June 30th, 2017 (early-summer). Seven of the 26 honey bee colonies were moved to the UNL ENREC agricultural field site location in July 2017 (mid-summer). Four additional colonies were moved to two privately-owned agricultural farms in spring of 2017. The Dixi Farm was a property growing roughly 2.02-hectares of alfalfa and was surrounded by adjacent corn and soybean fields. The second property (Goat Farm) was a 12.14-hectare organic vegetable and goat farm. Ten individual honey bees were collected from hive entrances at each of the four apiary sites (UNL Pollinator Garden, ENREC, Dixi Farm, and Goat Farm) in early-summer (May 1st-June 30th), mid-summer (July 1st-Aug 30th), and late summer (Sept 1st-30th). Due to colony losses and potential pesticide kills, honey bees were not collected from ENREC and the Goat Farm in late summer of 2017.

In 2018, colonies that successfully over-wintered in UNL Pollinator Garden and Dixi Farm were resampled. Colonies from the Goat Farm did not survive and were replaced with newly established colonies. Colonies that died at ENREC were not replace due to concerns over the previous year pesticide kill. Therefore, in 2018, only three of the four apiaries were resampled.

## Bee tissue dissection

Ten bees were collected for each target species per collection, and ten adult honey bees were collected from each colony for viral analysis totaling 1,643 specimens (*A. mellifera* (n = 1,180); *B. impatiens* (n = 154); *B. griseocollis* (n = 161); and *H. ligatus* (n = 148)) collected in 2017 and 2018. For tissue preparation, each bee was individually divided into three body regions (head, thorax, and abdomen). The wings, legs, head, and abdomen of bees were removed with scissors and forceps to reduce the likelihood of a false negative virus detection in PCR due to known inhibitory substances present in the compound eyes and guts of honey bees and insects [33]. The thoracic region of each bee was individually placed into sterile microcentrifuge tubes (1.5 ml) while the heads and abdomens were placed together into different microcentrifuge tubes. All samples were stored at -80°C until they were processed for RNA extraction.

## Pollen collection

To examine the role pollen plays in interspecific transmission of viruses, pollen loads collected by foraging honey bees were tested for the presence of viruses. Honey bee-collected pollen loads (50–100 mg) were obtained using pollen traps, which required returning bees to squeeze through a mesh screen that dislodged pollen off their pollen basket or corbicula [34]. Pollen traps were placed on three hives (located at the UNL Pollinator Garden) and activated for 24–48 hours per hive in early-mid-summer, and late-summer. The fresh pollen collected from returning foragers and retrieved by pollen traps was then processed and analyzed for the presence of the four target viruses.

## RNA extraction

Dissected bee thoraces and pollen samples were processed to extract total RNA and assessed for the presence of viral RNA. Bees were individually processed to improve detection of viruses in homogenate samples. Each sample was transferred to nuclease-free centrifuge tubes (1.5 ml, Thermo Fisher Scientific, Waltham, MA). Resistant 316 stainless steel ball (6.35 mm diameter) and TRI Reagent (ml per 50–100 mg of tissue; Invitrogen, Carlsbad, CA) were put into sample tubes to homogenize using the SPEX SamplePrep 2010 Geno/Grinder® at 1300 strokes for 1 min. The liquid was removed into a fresh centrifuge tube and allowed to settle for 5 min at room temperature. After settling, chloroform (0.2 ml per ml of TRIzol Reagent) was added and the tube was vigorously vortexed for 15 secs to mix. Samples were then left at room temperature for 15 min and centrifuged at 12,000 x g for 15 min. At this point, samples separated into three layers and the aqueous phase was transferred into a fresh tube. Isopropanol (0.5 ml per ml of TRIzol Reagent used) was added and gently mixed to the aqueous phase. Sample tubes were stored over night at -20 ˚C and then centrifuged at 12,000 x g for 10 min. After this process, a very small visible pellet containing RNA would form at base of each tube. RNA pellets were washed by adding 1 ml 75% ethanol and then centrifuging the tube at 7,500 x g for 5 min. The ethanol was poured off without touching the pellet at base of tubes and the pellet was allowed to air dry. RNA samples were further dissolved in DEPC-treated water in the presence of Ribonuclease Inhibitor (Invitrogen, Carlsbad, CA) and stored at -80 ˚C for further analysis [35]. The concentration of extracted RNA was estimated by measuring the absorbance of aliquots at 260nm of an aliquot of the final preparation at a Nanodrop 2000c (Thermo Fisher Scientific, Waltham, MA). Then, reverse transcription-PCR (RT-PCR) was used to detect the presence of target viruses.

## RT-PCR method

RT-PCR analysis was applied for detection of DWV, IABPV, BQCV, and SBV in the samples. Specific, publicly available primers were used for viruses that were as follows: DWV-F: 5′–ATCAGCGCTTAGTGGAGGAA–3′, DWV-R: 5′–TCGACAATTTTCGGACATCA–3′, IABPV-F: 5′–TTATGTGTCCAGAGACTGTATCCA–3′, IABPV-R: 5′–GCTCCTATTGCTCGGTTTTTCGGT–3′, BQCV-F: 5′–TGGTCAGCTCCCACTACCTTAAAC–3′, BQCV-R: 5′–GCAACAAGAAGAAACGTAAACCAC–3′, and SBV-F: 5′–GCTGAGGTAGGATCTTTGCGT–3′, SBV-R: 5′–TCATCATCTTCACCATCCGA–3′ (35).

## Target virus selection and assessment

Honey bees collected from colonies in 2016 were tested for the presence of IAPV, BQCV, DWV, KBV, SBV-1 and SBV-2 separately by the uniplex RT-PCR technique [35]. The Access Quick RT-PCR system (Promega, Madison, WI) was used according to the manufacturer's instructions. The reaction mixture contained: 25 μl of Access Quick TM Master Mix, 0.5 μl of Upstream Primer, 0.5 μl of Downstream Primer, 2.5 μl of RNA Template, 1l of AMV Reverse Transcriptase, and 500 ng total RNA in a total volume of 25 μl. The thermal cycling profiles were as follows: one cycle at 48 ˚C for 45 minutes for reverse transcription followed by 95˚C for 2min; up to 40 cycles at 95˚C for 30 sec., 55 ˚C for 1 min, and 68 ˚C for 2 min; 68 ˚C for 7 min. Negative and positive control samples (previously identified) were included in each RT-PCR trial [35]. Amplification products were analyzed together with a 100 bp ladder for size determination of PCR products through electrophoresis (1% agarose gel and 0.5 ug/ml ethidium bromide) and UV trans illumination [35] for visualization.

 Four of six viruses identified in 2016 honey bee samples became the target viruses (DWV, IAPV, BQCV, SBV) for 2017–2018 samples. One or more viruses may be detected within a

sample and may be defined as a mono-, di-, tri, or tetra-infection. To develop an assay that allowed simultaneous detection of different viruses in a single reaction, multiplex RT-PCR were used on samples of adult bee identified with infections of four viruses using uni-plex RT-PCR.

The multiplex RT-PCR protocols adapted from Chen et al. [35] were performed to determine whether bee and pollen samples contained DWV, IAPV, BQCV, and or SBV viral RNA by using a one-step RT-PCR kit (Promega, Madison, WI). The RT-PCR kit contained 1 x AMV/Tfl reaction buffer, 4 mM $MgSO_4$, 0.6 mM dNTP, 0.4 unit AMV reverse transcriptase, 0.4 unit Tfl DNA polymerase, 2 µg total RNA, 0.6 µM of each specific primer for DWV, IAPV, BQCV, and SBV for a total reaction volume of 50 µl. The cycling conditions consisted of one cycle at 48 ˚C for 45 min for reverse transcription, followed by one cycle at 94 ˚C for 2 min, 20 cycles of 94 ˚C for 30 s, 58 ˚C for 1 min, and 68 ˚C for 1 min, and 20 cycles of 94 ˚C for 30 s, 52 ˚C for 1 min and 68 ˚C for 1 min, followed by final elongation step at 68 ˚C for 10 min. PCR products were separated by electrophoresis in 1.8% agarose gel and visualized with Gel Doc™ XR + and ChemiDoc™ XRS + gel documentation systems using Image Lab™ software version 5.1 (Bio-Rad, Hercules, CA).

## Statistical analysis

**Proportion of infected bees across landscapes.** The proportion of bees (*A*, *mellifera*, *B. impatiens*, *B. griseocollis*, *H. ligatus*) that had viruses (BQCV, DWV, IAPV, and SBV) were compared across different landscape types (urban, agricultural, and open spaces) by dividing the number of virus and bees by total number of bees from that landscapes using Chi-Square test and Fisher's Exact test with the critical value for significance at alpha = 0.05. Fisher's Exact test was selected because it is robust for small sample sizes and zero-inflated response data. Fisher's Exact tests were performed using SAS or R version 3.5.1 [36].

**Community analysis of virus type.** To assess the relationship between virus types, bee host species, and season, a distance-based redundancy analysis (dbRDA) was performed [37]. This is an ordination method that allows analysis of non-Euclidian distances, such as Bray-Curtis distance that is appropriate for presence/absence data. First, pairwise dissimilarity between virus communities (presence/absence) from each environment was constructed using a zero-adjusted Bray-Curtis dissimilarity measure [38]. The vegan package in R (version 3.5.1) was used for calculation and analysis of dissimilarity [39]. The dissimilarity matrix was first ordinated using metric scaling with the function 'capscale' and the redundancy analysis summarizes variation with a model that identified explanatory variables best able to explain the variation. Explanatory variables evaluated were those associated with each site (landscape, host, season, and year) [40]. The optimal model for the constrained ordination was identified using the 'ordistep' function, which implements a forward–backward stepwise model selection procedure using permutation tests [41]. Negative eigenvalues were transformed to positive eigenvalues within the capscale analysis to discard imaginary dimensions where negative eigenvalues exist and only base models positive eigenvalues in real dimensional space [39]. Lastly, the 'adonis' function was applied to perform a permutation analysis of variance (PERMANOVA) on distance matrices to compare groups and determine if communities are significantly different (alpha = 0.05) [39].

**Multidimensional analysis of virus type.** To assess the relationship between virus types, bee host species, and season, non-metric multidimensional scaling (NMDS) was performed [37]. Differences among species (dissimilarity matrix) and the species distribution communities (presence/absence matrix) with a zero-adjusted Bray-Curtis dissimilarity metric were analyzed [38]. The vegan package in R (version 3.5.1) was used for the analysis of dissimilarity

[39]. To determine whether there is a significant difference between independent groups, Kruskal-Wallis H test was used [42]. When a significant difference was observed, the post-hoc Dunn's test was used to identify pairwise differences between groups [43]. The function metaMDS from vegan package was used for the NMDS analysis.

**Mixed virus infections in honey bee colonies.** To assess co-infection in different landscapes at honey bee apiaries, we compared the frequency of co-detection of viruses with the probability of co-detection of viruses, calculated as the product of the frequency of each virus. Specifically compared were the observed frequency of co-detection of viruses per honey bees and colonies collected from agricultural and urban landscapes in 2017 and 2018. The number of detections of each virus were categorized as to whether they occurred by themselves (mono-infection) or concurrently with one or more other viruses (co-detection: dual-, triple-, and tetra-infection). For each category, we calculated the magnitude (M) of difference between the observed frequency of co-detection and expected frequency of co-detection. Magnitude (M) values not significantly different from 1.0 suggest no difference in the ratio of observed to expect frequency of co-occurring viruses, however, values significantly greater than 1.0 suggest co-infections occurring at a higher frequency than expected by chance and those significantly lower than 1.0 suggest co-infections are less frequent than expected.

## Result

### Virus prevalence in foraging bees

A total of 1,643 bees (*A. mellifera* = 1,180, *B. impatiens* = 154, *B. griseocollis* = 161, and *H. ligatus* = 148) were collected from agricultural (n = 385), urban (n = 1140), and open sites (n = 154) during the summer in 2017 and 2018. Bees were analyzed for the presence of DWV, IAPV, BQCV, and SBV and compared across landscape types and season. All four viruses were detected in honey bees (*A. mellifera)* collected from agricultural sites, particularly within late-summer collections. However, DWV and IAPV were detected mostly in sweat bees (*H. ligatus*; 22.4% and 20.7%, respectively) collected from urban and open sites in both early-mid and late summer collections (Table 1). The frequency of positive detections among the four target bee species were separately analyzed by landscape type and across season and year (S1 Fig).

To assess the potential role of bee management on viruses, bee species were further grouped by whether they were actively "managed" species or are "unmanaged" wild species. Honey bees (*A. mellifera*) and one bumble bee species (*B. impatiens*) are commercially managed for pollination and or honey production. The Union Roadside (open space) site had 6–8 commercial *B. impatiens* colonies stocked in the area, however unmanaged wild *B. impatiens* were also be found readily. Therefore, *B. impatiens* was grouped or categorized, in this study, as a "managed" bee species. Managed bee colonies are typically stocked at higher densities then natural densities of wild species (*B. griseocollis* and *H. ligatus)* and thus may play a greater role of viral persistence and transmission. Comparisons between managed and unmanaged bee groups collected from three different landscapes, indicated there were significant differences in the number of positive detections of DWV in unmanaged compared to managed bees but only from urban sites in 2018 (p = 0.012, p ≤ 0.001 for early-mid and late summer; Tables 2 and 3). Differences in the number of positive IAPV detections in early-mid (open and urban) and late summer (urban) samples collected in 2018 were also observed among managed and unmanaged bees (p = 0.001, p < 0.0001 for early-mid and late summer, respectively; Tables 2 and 3). Data indicate higher prevalence of IAPV (more frequent detections) in unmanaged bees than managed bees in both urban and open sites for early-mid and late summer (Table 2). However, no significant differences were observed in BQCV and SBV detections between managed and

**Table 1. The percent of individual bees (*Apis mellifera*, *Bombus impatiens*, *Bombus griseocollis*, *Halictus ligatus*) (n/total bees) with detectable levels of viruses (BQCV, DWV, IAPV, and SBV) across different landscape types (urban, agricultural, and open spaces) in 2017 and 2018.**

| Virus Group | Bee Species | Agriculture | | | | Urban | | | | Open | | | |
|---|---|---|---|---|---|---|---|---|---|---|---|---|---|
| | | 2017 | | 2018 | | 2017 | | 2018 | | 2017 | | 2018 | |
| | | Early-mid summer | Late summer | Early-mid summer | Late summer | Early-mid summer | Late summer | Early-mid summer | Late summer | Early-mid summer | Late summer | Early-mid summer | Late summer |
| DWV | *Apis mellifera* | 0.7 (1/150) | 73.3 (22/30) | 18.8 (15/80) | 10.0 (4/40) | 5.2 (23/440) | 5.0 (7/140) | 5.5 (11/200) | 7.0 (7/100) | NA | NA | NA | NA |
| | *Bombus impatiens* | 0 (0/1) | 0 (0/3) | 0 (0/9) | 6.3 (1/16) | 0 (0/13) | 0 (0/22) | 33.3 (2/6) | 9.5 (4/42) | 0 (0/6) | 0 (0/10) | 0 (0/10) | 14.8 (1/13) |
| | *Bombus griseocollis* | 0 (0/10) | NA | 13.3 (2/15) | 25.0 (1/4) | 0 (0/19) | 0 (0/8) | 3.0 (1/33) | 0 (0/4) | 10.0 (1/10) | NA | 7.7 (3/39) | 14.9 (2/10) |
| | *Halictus ligatus* | 0 (0/4) | 100 (1/1) | 38.1 (8/21) | 0 (0/1) | 18.5 (5/27) | 0 (0/4) | 40.9 (9/22) | 88.9 (8/9) | 21.1 (4/19) | NA | 43.8 (14/32) | 60.0 (3/5) |
| IAPV | *Apis mellifera* | 0 (0/150) | 23.3 (7/30) | 1.3 (1/80) | 10.0 (4/40) | 1.1 (5/440) | 1.4 (2/140) | 0.5 (1/200) | 3 (1/100) | NA | NA | NA | NA |
| | *Bombus impatiens* | 0 (0/1) | 0 (0/3) | 0 (0/9) | 0 (0/16) | 0 (0/13) | 0 (0/22) | 0 (0/6) | 0 (0/42) | 0 (0/6) | 0 (0/10) | 0 (0/10) | 0 (0/13) |
| | *Bombus griseocollis* | 0 (0/10) | NA | 0 (0/15) | 0 (0/4) | 0 (0/19) | 0 (0/8) | 0 (0/33) | 0 (0/4) | 0 (0/10) | NA | 2.6 (1/39) | 20 (2/10) |
| | *Halictus ligatus* | 0 (0/4) | 0 (0/1) | 14.3 (3/21) | 0 (0/1) | 7.4 (2/27) | 0 (0/4) | 36.4 (8/22) | 88.9 (8/9) | 0 (0/19) | NA | 21.9 (7/32) | 80 (4/5) |
| SBV | *Apis mellifera* | 0 (0/150) | 23.3 (7/30) | 1.3 (1/80) | 7.5 (3/40) | 1.1 (5/440) | 0 (0/140) | 0.5 (1/200) | 5 (5/100) | NA | NA | NA | NA |
| | *Bombus impatiens* | 0 (0/1) | 0 (0/3) | 0 (0/9) | 0 (0/16) | 0 (0/13) | 0 (0/22) | 0 (0/6) | 4.8 (2/42) | 0 (0/6) | 0 (0/10) | 0 (0/10) | 0 (0/13) |
| | *Bombus griseocollis* | 0 (0/10) | NA | 6.7 (1/15) | 50 (2/4) | 0 (0/19) | 0 (0/8) | 0 (0/33) | 0 (0/4) | 0 (0/10) | NA | 0 (0/39) | 0 (0/10) |
| | *Halictus ligatus* | 0 (0/4) | 0 (0/1) | 0 (0/21) | 0 (0/1) | 0 (0/27) | 0 (0/4) | 0 (0/22) | 0 (0/9) | 0 (0/19) | NA | 0 (0/32) | 20 (1/5) |
| BQCV | *Apis mellifera* | 0 (0/150) | 16.7 (5/30) | 0 (0/80) | 0 (0/40) | 0.9 (4/440) | 0 (0/140) | 0 (0/200) | 1 (1/100) | NA | NA | NA | NA |
| | *Bombus impatiens* | 0 (0/1) | 0 (0/3) | 0 (0/9) | 0 (0/16) | 0 (0/13) | 0 (0/22) | 0 (0/6) | 0 (0/42) | 0 (0/6) | 0 (0/10) | 0 (0/10) | 0 (0/13) |
| | *Bombus griseocollis* | 0 (0/10) | NA | 0 (0/15) | 0 (0/4) | 0 (0/19) | 0 (0/8) | 0 (0/33) | 0 (0/4) | 0 (0/10) | NA | 0 (0/39) | 10 (1/10) |
| | *Halictus ligatus* | 0 (0/4) | 0 (0/1) | 0 (0/21) | 0 (0/1) | 0 (0/27) | 0 (0/4) | 0 (0/22) | 0 (0/9) | 0 (0/19) | NA | 0 (0/32) | 0 (0/5) |

Percent was determined by Fisher's Exact Test and denoted by "*"at alpha = 0.05. For Agriculture, Urban, and Open landscapes, the names of viruses are abbreviated as follow: **DWV**, Deformed wing virus; **BQCV**, Black queen cell virus; **IAPV**, Israeli acute paralysis virus; **SBV**, Sacbrood virus. NA denotes no bees found or collected. Four bee species were collected in early-mid summer (May 1st-July 30th) and late summer (August 1st-Sept 30th).

unmanaged bees collected in agriculture sites (BQCV; p> 0.0001 for both early-mid and late summer in 2018 and SBV; p = 0.494, p = 0.049 early-mid and late summer in 2018, respectively; Table 3).

Results for bees collected in agricultural sites (n = 385) indicate no effect by year (p = 0.065) and a significant correlation between the prevalence of DWV- and SBV-infected foragers and late summer collections for both years (Fisher Exact test; p = 0.027 and p = 0.042, respectively; Table 3). Bees collected from urban sites (n = 1140) had more detections of DWV and IAPV in both early-mid and late summer samples particularly in 2018 (p < 0.001; Tables 1 and 3).

Within natural or open sites, DWV detections were significantly different across wild bees (no honey bees caught), but only in early season samples in 2018 [DWV p = 0.0003 (early-mid)]. IAPV detections were also significantly different among wild bees in 2018 and

**Table 2. The percent of managed (*Apis mellifera & Bombus impatiens*) & unmanaged wild bees (*Bombus giseocollis & Halictus ligatus*) with detectable levels of viruses (Deformed wing virus (DWV), Israeli acute paralysis virus (IAPV), Sacbrood virus (SBV), and Black queen cell virus (BQCV)) across different landscape types (Agricultural (n = 385), Urban (n = 1140), and Open sites (n = 154)) during 2017 and 2018.**

| Virus Group | Bee Group | Agriculture | | | | Urban | | | | Open | | | |
|---|---|---|---|---|---|---|---|---|---|---|---|---|---|
| | | 2017 | | 2018 | | 2017 | | 2018 | | 2017 | | 2018 | |
| | | Early-mid Summer | Late Summer | Early-mid Summer | Late Summer | Early-mid Summer | Late Summer | Early-mid Summer | Late Summer | Early-mid Summer | Late Summer | Early-mid Summer | Late Summer |
| DWV | Managed bees | 0.7 | 66.7 | 16.9 | 8.9 | 5.1 | 4.3 | 6.3 | 7.8 | 0 | 0 | 0 | 7.7 |
| | Unmanaged bees | 0 | 100 | 27.8 | 20 | 1 | 0 | 18.2* | 61.5* | 17.2 | 0 | 23.9 | 33.3 |
| IAPV | Managed bees | 0 | 21.2 | 1.1 | 7.1 | 1.1 | 1.2 | 0.5 | 2.1 | 0 | 0 | 0 | 0 |
| | Unmanaged bees | 0 | 0 | 8.3 | 0 | 4.4 | 0 | 14.6* | 61.5* | 0 | 0 | 11.3 | 40* |
| SBV | Managed bees | 0 | 21.2 | 1.1 | 5.4 | 1.1 | 0 | 0.5 | 4.9 | 0 | 0 | 0 | 0 |
| | Unmanaged bees | 0 | 0 | 2.8 | 40 | 0 | 0 | 0 | 0 | 0 | 0 | 0 | 6.7 |
| BQCV | Managed bees | 0 | 15.2 | 0 | 0 | 0.9 | 0 | 0 | 0.7 | 0 | 0 | 0 | 0 |
| | Unmanaged bees | 0 | 0 | 0 | 0 | 0 | 0 | 0 | 0 | 0 | 0 | 0 | 6.7 |

Significant differences in virus prevalence were determined using Fisher's Exact Test and at alpha < 0.05. Outlined squares denotes significant differences observed among all four bee species (solid line p<0.05, dotted line p = 0.05)) while "*" denotes where significantly higher viral detections (p<0.05) were observed in either managed or unmanaged bee categories.

differences were observed for early-mid and late summer samples (p = 0.0183 and p = 0.0012, respectively; Table 3). Positive detections of viruses from bees collected in natural landscapes (5.3%) or open spaces (4.4%) were similar to detection rates in bees from urban sites (5.8%), and yielded more positive detections of DWV (11.1%) and IAPV (6.5%) and few detections of BQCV (0.6%) and SBV (2.5%). Higher detections of DWV (37.8%) and IAPV (40.2%) in sweat bees, *H. ligatus*, accounted for the significant differences observed across bee species, but only in late season in 2018 (p = 0.0012; Tables 1 and 3). Positive detections of viruses showed differences among bee species (p = 0.020) and season (p = 0.005), but no differences among landscape type (p = 0.320) and year (p = 0.070).

## Dissimilarities among bee species, virus type, and season

The occurrence and dissimilarities of viruses in each sample relative to all other samples was visualized by using Bray-Curtis analysis. Results indicated that bee species ($R^2 = 0.089$, p < 0.001) and season ($R^2 = 0.036$, p < 0.001) were significant main factors that accounted for the variance observed (Fig 1 and Table 4). Landscape type and year, however, were not significant factors in viral prevalence (Table 4, p = 0.290 and p = 0.065, respectively).

Relationships between virus types showed DWV was the most prevalent and commonly detected virus across all bee species, landscape types, and seasons. The high prevalence of DWV in bees positively correlated with IAPV, but not with BQCV and SBV (Figs 1 and 2). Few viruses were detected in pollen samples collected from the legs of foraging honey bees (trapped pollen, DWV n = 1/9, rate = 11.1% in early-mid in 2018), in-hive pollen stores (DWV n = 1/6 and BQCV n = 1/6, rate = 16.6% and DWV n = 1/2, rate = 50% in early-mid and late

**Table 3. Descriptive statistics for analyses on positive detection rates of four common honey bee viruses (Deformed wing virus (DWV), Israeli acute paralysis virus (IAPV), Sacbrood virus (SBV), and Black queen cell virus (BQCV)) in managed and wild bees collected from three landscapes types (Agricultural or "Ag" (n = 385), Urban (n = 1140), and Open sites (n = 154)) across two years, 2017 and 2018.**

| Ag | Early-mid Summer 2017 | | | | Late Summer 2017 | | | | Early-mid Summer 2018 | | | | Late Summer 2018 | | | |
|---|---|---|---|---|---|---|---|---|---|---|---|---|---|---|---|---|
| Virus Group | Df | N | X2 | p | df | N | X2 | p | df | N | X2 | p | df | N | X2 | p |
| BQCV | 0 | 165 | 0 | 0 | 2 | 34 | 0.7816 | 0.6765 | 0 | 125 | 0 | 0 | 0 | 61 | 0 | 0 |
| DWV | 3 | 165 | 0.1006 | 0.9918 | 2 | 34 | 7.1942 | 0.0274* | 3 | 125 | 7.0424 | 0.0897 | 3 | 61 | 1.3794 | 0.6129 |
| SBV | 0 | 165 | 0 | 0 | 2 | 34 | 1.1753 | 0.5556 | 3 | 125 | 2.9959 | 0.3755 | 3 | 61 | 10.832 | 0.0427* |
| IAPV | 0 | 165 | 0 | 0 | 2 | 34 | 1.1753 | 0.5556 | 3 | 125 | 10.1069 | 0.0493 | 3 | 61 | 2.2472 | 0.5177 |
| **Urban** | **Early-mid Summer 2017** | | | | **Late Summer 2017** | | | | **Early-mid Summer 2018** | | | | **Late Summer 2018** | | | |
| Virus Group | Df | N | X2 | p | df | N | X2 | p | df | N | X2 | p | df | N | X2 | p |
| BQCV | 3 | 499 | 0.5407 | 0.9099 | 0 | 174 | 0 | 0 | 0 | 261 | 0 | 0 | 3 | 155 | 0.5536 | 1 |
| DWV | 3 | 499 | 10.517 | 0.0505 | 3 | 174 | 1.7713 | 0.7566 | 3 | 261 | 36.7979 | < .0001*UM | 3 | 155 | 52.5596 | < .0001*UM |
| SBV | 3 | 499 | 0.6772 | 1 | 0 | 174 | 0 | 0 | 3 | 261 | 0.3062 | 1 | 3 | 155 | 0.675 | 1 |
| IAPV | 3 | 499 | 7.7195 | 0.1487 | 3 | 174 | 0.4914 | 1 | 3 | 261 | 78.2054 | < .0001*UM | 3 | 155 | 97.3811 | < .0001*UM |
| **Open** | **Early-mid Summer 2017** | | | | **Late Summer 2017** | | | | **Early-mid Summer 2018** | | | | **Late Summer 2018** | | | |
| Virus Group | Df | N | X2 | p | df | N | X2 | p | df | N | X2 | p | df | N | X2 | p |
| BQCV | 0 | 35 | 0 | 0 | 0 | 10 | 0 | 0 | 0 | 81 | 0 | 0 | 2 | 28 | 1.8667 | 0.5357 |
| DWV | 2 | 35 | 1.8605 | 0.3944 | 0 | 10 | 0 | 0 | 2 | 81 | 16.8118 | 0.0003* | 2 | 28 | 5.8871 | 0.0524 |
| SBV | 0 | 35 | 0 | 0 | 0 | 10 | 0 | 0 | 2 | 81 | 0 | 0 | 2 | 28 | 4.7704 | 0.1786 |
| IAPV | 0 | 35 | 0 | 0 | 0 | 10 | 0 | 0 | 2 | 81 | 8.6143 | 0.0183* | 2 | 28 | 13.7455 | 0.0012*UM |

Significant differences in virus prevalence among the four bee species were determined using the Fisher's Exact Test and denoted by "*" at alpha < 0.05. Significantly higher detections of viruses observed in managed (*M) or unmanaged (*UM) bee categories are also shown at alpha < 0.05.

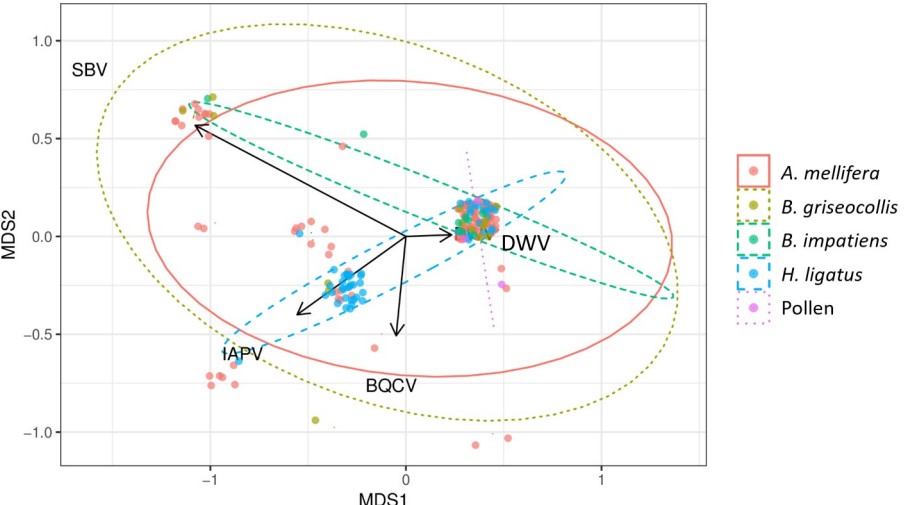

**Fig 1. Results from the non-metric multidimensional scaling (NMDS) plot showed the prevalence of viruses in each sample relative to all other samples.** To assess the differences between species (dissimilarity matrix) and the species distribution communities (presence/absence matrix), Bray-Curtis dissimilarity were calculated and visualized using NMDS with an associated stress value of (0.08988) in R (version 3.5.1). MDS1 and MDS2 are the first two axes that show the most variation from the multidimensional scaling coordinates, with the respective amount of variation associated with each axis. Black arrows are eigenvectors that correspond with DWV, IAPV, BQCV, and SBV virus detections. Individual infections are denoted by solid dots. Samples with more similar virus compositions are closer in proximity to each other and size of points indicate frequency of positive detections. Virus compositions of bee groups were defined by clusters (denoted by colored circles i.e., *Apis mellifera*—orange, *Bombus griseocollis*—yellow, *Bombus impatiens*—green, *Halictus ligatus*—blue, or Pollen—purple). ** Significance from the 'anova.cca' test indicated differences among species with 1,000 permutations and denoted by p < 0.001.

**Table 4. Summary statistics from the Bray-Curtis analysis of dissimilarity illustrating the variance and significance in virus prevalence across bee species, season, and year.**

| | Bray-Curtis | |
| --- | --- | --- |
| | $R^2$ | *P*-value |
| Species | 0.08988 | **0.001** |
| Season times | 0.03699 | **0.001** |
| Year | 0.01154 | 0.065 |

The variances ($r^2$) and *P*- values were calculated using analysis of dissimilarity (adonis)

Bold *P*-values indicate significant comparisons from adonis test (*P* = 0.05).

summer, respectively, in 2017, S2 Fig). In general, viral prevalence in bees increased from early-mid to late summer collections in one or both years (Tables 1 and 2, and Fig 1). Viral prevalence for DWV averaged 0.084 (SD ± 0.27), IAPV averaged 0.023 (SD ± 0.15), BQCV averaged 0.004 (SD ± 0.06), and SBV averaged 0.007 (SD ± 0.08) in early-mid summer and DWV averaged 0.133 (SD ± 0.33), IAPV averaged 0.063 (SD ± 0.24), BQCV averaged 0.015 (SD ± 0.12), and SBV averaged 0.042 (SD ± 0.20) in late summer (average of viral prevalence in bees ± standard deviation; Table 5).

## Differences between groups

The differences in virus prevalence in bee samples were quantified by using a Bray-Curtis dissimilarity matrix. Kruskal-Wallis test ($\alpha$ = 0.05) used to compare viruses for bee groups and season times revealed that DWV and IAPV were significant in bee groups and DWV, IAPV,

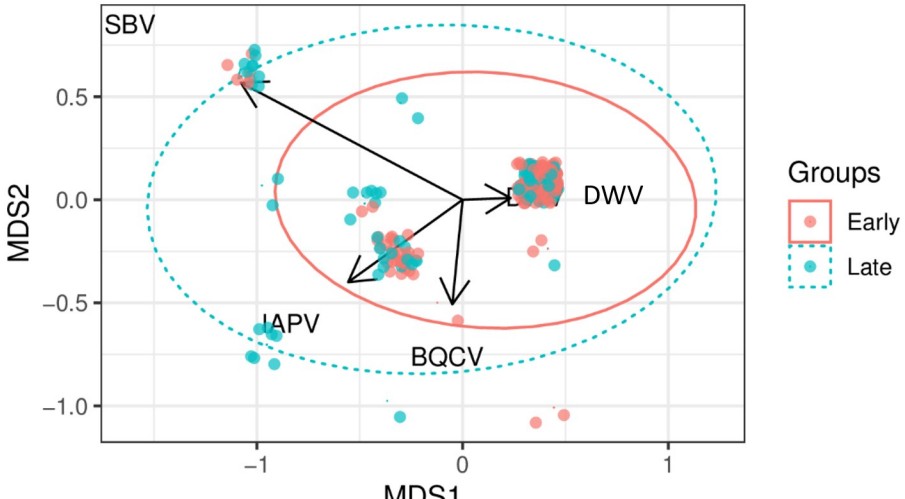

**Fig 2. Results from the non-metric multidimensional scaling (NMDS) plot showed the prevalence of viruses in each sample relative to all other samples in early and late summer.** To assess the differences between species (dissimilarity matrix) and the species distribution communities (presence/absence matrix), Bray-Curtis dissimilarity were used on NMDS with an associated stress value of (0.0369) in R (version 3.5.1). MDS1 and MDS2 are the first two axes from the multidimensional scaling coordinates, with the respective amount of variation in the Bray–Curtis. Black arrows correspond with DWV, IAPV, BQCV, and SBV virus detections. Individual infections are denoted by solid dots. Samples with more similar virus compositions are closer in proximity and size of points indicate frequency of positive detections. Virus compositions of experiment time were defined by clusters (denoted by colored circles i.e., Early summer––orange, or Late summer––green). ** Significance from the 'anova.cca' test was used to indicate differences among season with 1,000 permutations and denoted by p < 0.001.

**Table 5. The mean number (± s.d.) of positive detections of viruses present in sampled bee species by landscape types, season, and year.**

| by landscapes | DWV | IAPV | BQCV | SBV |
|---|---|---|---|---|
| Urban | 0.070 ± 0.256 | 0.025 ± 0.158 | 0.005 ± 0.072 | 0.011 ± 0.106 |
| Agriculture | 0.143 ± 0.350 | 0.039 ± 0.194 | 0.013 ± 0.113 | 0.036 ± 0.187 |
| Open | 0.182 ± 0.387 | 0.091 ± 0.288 | 0.091 ± 0.081 | 0.006 ± 0.081 |
| by season | | | | |
| Early-mid | 0.084 ± 0.278 | 0.023 ± 0.151 | 0.004 ± 0.064 | 0.007 ± 0.081 |
| Late | 0.130 ± 0.336 | 0.063 ± 0.243 | 0.015 ± 0.120 | 0.042 ± 0.200 |
| by year | | | | |
| 2017 | 0.070 ± 0.255 | 0.017 ± 0.129 | 0.011 ± 0.102 | 0.013 ± 0.112 |
| 2018 | 0.133 ± 0.339 | 0.133 ± 0.233 | 0.003 ± 0.052 | 0.022 ± 0.146 |

Mean number of virus prevalence in bee species ± standard deviation

SBV, and BQCV is significant in both early-mid and late summer ($p < 0.05$; Table 6). Prevalence of DWV and IAPV were significantly different between *Halictus ligatus* and all other bee species (*A. mellifera*, *B. impatiens*, and *B. griseocollis*), and different within pollen by using Dunn's test ($p < 0.001$). Due to a limitation in season sample size (df = 1), Dunn's test was not applied to compare viral prevalence by season (Table 7).

**Table 6. Analysis of differences between bee groups by using Kruskal-Wallis test.**

| Each virus for bee species | Ch2 | df | p-value |
|---|---|---|---|
| DWV | 124.81 | 4 | 2.20E-16 |
| IAPV | 166.79 | 4 | 2.20E-16 |
| SBV | 2.1832 | 4 | 0.7021 |
| BQCV | 3.5445 | 4 | 0.4711 |
| **Each virus for season** | **Ch2** | **df** | **p-value** |
| DWV | 8.1097 | 1 | 0.004403 |
| IAPV | 15.944 | 1 | 6.53E-05 |
| SBV | 25.79 | 1 | 3.81E-07 |
| BQCV | 5.2901 | 1 | 0.02145 |

Names of viruses are abbreviated as follow: **DWV**, Deformed wing virus; **BQCV**, Black queen cell virus; **IAPV**, Israeli acute paralysis virus; **SBV**, Sacbrood virus.

**Table 7. Analysis of differences between bee groups by using Dunn's test (significant species for only significant viruses).**

| Viruses | Bee groups | Dunn's pairwise z test | p-value |
|---|---|---|---|
| DWV | *H.ligatus-Apis mellifera* | -10.83383 | 0.0000* |
| | *H.ligatus-Bombus griseocollis* | -8.51771 | 0.0000* |
| | *H.ligatus-Bombus impatiens* | -8.875963 | 0.0000* |
| | pollen-*Halictus ligatus* | 6.217939 | 0.0000* |
| IAPV | *H.ligatus-Apis mellifera* | -12.51574 | 0.0000* |
| | *H.ligatus-Bombus griseocollis* | -9.476271 | 0.0000* |
| | *H.ligatus-Bombus impatiens* | -10.39021 | 0.0000* |
| | pollen-*Halictus ligatus* | 7.420597 | 0.0000* |

The names of viruses are abbreviated as follow: **DWV**, Deformed wing virus; **IAPV**, Israeli acute paralysis virus.

## Mixed virus infections in honey bee colonies

From the urban apiary, 440 individual honey bees were collected from 44 different colonies in early-mid and late summer of 2017 and 2018 (Table 4). The percent of colonies that had detectable viruses increased in late season samples for both years. In 2017, the percent of colonies with one target virus present (mono-infection of DWV, IAPV, BQCV, or SBV) ranged between 5.2–18.1% and 5–42.9% in early and late summer collections, respectively. In 2018, the percent range of infected colonies dropped slightly to 5.5–30% and 7–40%, respectively. Less than 10% of colonies had two or move viruses present (di-, tri, tetra-infections) in both years (Table 8).

Within the agricultural sites, 30–150 honey bees were randomly collected from 3–15 colonies at each of the four apiaries. High rates of mono-, di-, tri-, and tetra-infections were found in bees collected from colonies in agricultural sites (15 total colonies) compared to bees collected from colonies in urban sites (44 total colonies). The infection rate of bees increased in

**Table 8. Observed frequency compared with expected frequency of viruses per honey bees and colonies collected from urban landscapes in 2017 and 2018.**

| Season | Type of Infection | Viruses detected | % of bees infected (n/total bees) | % of colonies infected (n/total colonies) | Rate expected per bee (%) | M |
|---|---|---|---|---|---|---|
| early-mid summer 2017 | Mono-infection | DWV | 5.2% (23/440) | 18.1% (8/44) | | |
| | | IAPV | 1.1% (5/440) | 4.5% (2/44) | | |
| | | SBV | 1.1% (5/440) | 9% (4/44) | | |
| | | BQPV | 1% (4/440) | 6.9% (3/44) | | |
| | Di-infection | DWV/IAPV | 1.1% (5/440) | 4.5% (2/44) | 0.06 | 18X |
| | | DWV/BQCV | 0.5% (2/440) | 4.5% (2/44) | 0.05 | 10X |
| | | DWV/SBV | 0.2% (1/440) | 2.3% (1/44) | 0.06 | 3X |
| | | IAPV/SBV | 0.2% (1/440) | 2.3% (1/44) | 0.01 | 20X |
| | | IAPV/BQCV | 0.2% (1/440) | 2.3% (1/44) | 0.01 | 20X |
| | Tetra-infection | DWV/IAPV/SBV | 0.2% (1/440) | 2.3% (1/44) | 0.01 | 20X |
| | | DWV/IAPV/ BQCV | 0.2% (1/440) | 2.3% (1/44) | 0.01 | 20X |
| late summer 2017 | Mono-infection | DWV | 5% (7/140) | 42.9% (6/14) | | |
| | | IAPV | 1.4% (2/140) | 14.3% (2/14) | | |
| | Di-infection | DWV/IAPV | 1.4% (1/140) | 7.1% (1/14) | 0.07 | 20X |
| early-mid summer 2018 | Mono-infection | DWV | 5.5% (11/200) | 30% (6/20) | | |
| | | IAPV | 0.5% (1/200) | 5% (1/20) | | |
| | | SBV | 0.5% (1/200) | 5% (1/20) | | |
| | Di-infection | DWV/IAPV | 0.5% (1/200) | 5% (1/20) | 0.03 | 17X |
| | | DWV/SBV | 0.5% (1/200) | 5% (1/20) | 0.03 | 17X |
| | | IAPV/SBV | 0.5% (1/200) | 5% (1/20) | 0.002 | 250X |
| | Tetra-infection | DWV/SBV/ IAPV | 0.5% (1/200) | 5% (1/20) | 0.0001 | 5,000X |
| late summer 2018 | Mono-infection | DWV | 7% (7/100) | 40% (4/10) | | |
| | | IAPV | 3% (3/100) | 20% (2/10) | | |
| | | SBV | 5% (5/100) | 30% (3/10) | | |
| | | BQPV | 1% (1/100) | 10% (1/10) | | |
| | Di-infection | DWV/BQCV | 1% (1/100) | 10% (1/10) | 0.07 | 14X |

The number of viruses positively detected (n) and the percentage of infected bees and colonies are shown. The total detection of each virus are divided into whether they occurred by themselves (mono-infection) or concurrently with one or more other viruses (co-detection: di-, tri- & tetra-infection). Expected frequency of co-detection were calculated as prevalence of first virus x prevalence of second virus. Magnitude (M) illustrates higher magnitude of fold change in observed co-infections than what is expected by chance.

**Table 9. Observed frequency compared with expected frequency of viruses per honey bees and colonies collected from agricultural landscapes in 2017 and 2018.**

| Season | Type of Infection | Viruses detected | % of bees infected (n/total bees) | % of colonies infected (n/total colonies) | Rate expected per bee (%) | M |
|---|---|---|---|---|---|---|
| early-mid summer 2017 | Mono-infection | DWV | 0.7%(1/150) | 6.6% (1/15) | | |
| late summer 2017 | Mono-infection | DWV | 73.3% (22/30) | 100% (3/3) | | |
| | | IAPV | 23.3% (7/30) | 66.6% (2/3) | | |
| | | SBV | 23.3% (7/30) | 66.6% (2/3) | | |
| | | BQPV | 16.6% (5/30) | 33.3% (1/3) | | |
| | Di-infection | DWV/IAPV | 23.3% (7/30) | 66.6% (2/3) | 17.1 | 1X |
| | | DWV/BQCV | 16.6% (5/30) | 33.3% (1/3) | 12.2 | 1X |
| | | DWV/SBV | 23.3% (7/30) | 66.6% (2/3) | 17.1 | 1X |
| | | BQCV/IAPV | 16.6% (5/30) | 33.3% (1/3) | 3.9 | 4X |
| | | BQCV/SBV | 16.6% (5/30) | 33.3% (1/3) | 3.9 | 4X |
| | | IAPV/SBV | 20% (6/30) | 33.3% (1/3) | 5.4 | 4X |
| | Tri-infection | DWV/BQCV/IAPV | 16.6% (5/30) | 33.3% (1/3) | 2.8 | 6X |
| | | DWV/IAPV/SBV | 20% (6/30) | 33.3% (1/3) | 4 | 5X |
| | | IAPV/SBV/BQCV | 16.6% (5/30) | 33.3% (1/3) | 0.9 | 18X |
| | | DWV/BQCV/SBV | 16.6% (5/30) | 33.3% (1/3) | 2.8 | 6X |
| | Tetra-infection | DWV/BQCV/SBV/IAPV | 16.6% (5/30) | 33.3% (1/3) | 0.7 | 23X |
| early-mid summer 2018 | Mono-infection | DWV | 18.7%(15/80) | 25% (2/8) | | |
| | | IAPV | 1.2% (1/80) | 25% (2/8) | | |
| | | SBV | 1.2% (1/80) | 12.5%(1/8) | | |
| | Di-infection | DWV/IAPV | 1.2% (1/80) | 12.5%(1/8) | 0.2 | 6X |
| late summer 2018 | Mono-infection | DWV | 10% (4/40) | 50% (2/4) | | |
| | | IAPV | 10% (4/40) | 50% (2/4) | | |
| | | SBV | 7.5% (3/40) | 25% (1/4) | | |
| | Di-infection | IAPV/SBV | 5% (2/40) | 25% (1/4) | 0.7 | 7X |

The number of viruses positively detected (n) and the percentage of infected bees and colonies are shown. The total detection of each virus is divided into whether they occurred by themselves (mono-infection) or concurrently with one or more other viruses (co-detection: di-, tri- & tetra-infection). Expected frequency of co-detection were calculated as prevalence of first virus x prevalence of second virus. Magnitude (M) illustrates higher magnitude of fold change in observed co-infections than what is expected by chance.

late season samples, which was consistent with temporal variation in rates from urban sites (Table 9). No apiaries were in close proximity to open space sites; therefore, no honey bees were collected.

DWV was the most prevalent virus detected among colonies, throughout the season, and in different landscapes with 5.2% and 5.5% at early summer of 2017 and 2018, 5%, and 7% at late summer of 2017 and 2018 in urban and 0.7% and 18.7% at early summer of 2017 and 2018, 73.3%, and 10% at late summer of 2017 and 2018 in agriculture (Tables 8 and 9). Multiple viruses-infections were observed at higher rates than what we would expect by chance and indicates an association between virus types. Calculations of expected frequency of co-detections suggest a relationship between the presence of DWV and increased likelihood of co-infection by other viruses with the higher observed dual- and triple-infection of DWV. For example, from Table 5, dual-infection of DWV with IAPV, BQCV, and SBV resulted in 18, 10, and 3-fold and triple-infection of DWV with three viruses resulted in 20-fold higher magnitude than expected infection base on frequency at early summer of 2017 in urban, respectively.

## Discussion

### Viral differentiation among bee species

In this study, four species of bees collected from different landscapes were analyzed for the presence of viruses (DWV, IAPV, BQCV, SBV) commonly detected in honey bees to improve our understanding of the distribution of viruses among bee communities in agricultural, urban, and natural open landscapes in eastern Nebraska. Recently, studies indicate multiple strains or variants of Deformed wing virus (DWV) exists which include type A (DWV; [44] and Kakugo virus; [45]), type B (Varroa destructor virus-1 (VDV-1); [46, 47]), and type C (phylogenetically distinct from type A and B; [47, 48]). Variants of DWV were detected on both honey bees and Varroa mite vectors; however, variants appear to have different effects on honey bee colonies [48, 49]. Type A is more prevalent and associated with colony death than compared to type B which is pathogenic at the individual bee level and is associated with a mite-borne virus (VDV-1; *Varroa destructor virus*-1) [49, 50]. DWV-C appears to be associated with DWV-A and has been indicated as a contributing factor in overwintering losses of colonies [50], however, there is limited knowledge available on the potential effects of DWV variants, especially DWV type C, on honey bee health. In this study, DWV variants were not assessed, however, DWV was the most frequently detected virus (11.1%) in all bee species, landscapes, seasonal times, and years. IAPV was the second most common virus (6.5%) found but was not found in all bee species, seasonal times, or years. BQCV (0.6%) and SBV (2.5%) were less commonly found yet were detected in honey bees collected from agricultural sites and bumble bees (*B. griseocollis*) from both agricultural and open sites. Detections of viruses common to honey bees in other hosts supports the hypothesis of inter-species transmission within pollinator communities and has been reported in several previous studies. However, it's unclear whether these viruses originated from honey bees or if honey bees became a more suitable host given the social dynamics of nestmates and high population densities within colonies that allow viruses to remain and persist [20, 28, 51–53]. Surprisingly, the significant differences observed between bee species for both DWV and IAPV appear to be driven by higher positive detections in wild sweat bees, *H. ligatus* (41% and 88.9% infected with DWV, and 36.4% and 88.9% infected with IAPV), which were collected from the urban site in early and late summer of 2018, respectively (Table 1). DWV and IAPV are varroa mite-mediated viruses and would be expected to appear at higher rates in honey bees, the only known host of varroa mites [54–57].

Less virulent viruses may be persistent and slowly spread within colonies when infected brood are able to survive through adulthood, thus providing mite-vectors with hosts capable of transporting mites and their associated viruses to uninfected bees and colonies [58]. High varroa mite populations in honey bee colonies is known to increase viral prevalence especially DWV within and across colonies [59–61]. High prevalence of DWV and IAPV in honey bees, bumble bees, and sweat bees, reaffirms that wild bee communities have high honey bee pathogens prevalence, however, little remains known about the impact of these viruses on wild bees [18]. Differences in tongue length and foraging behavior may be a plausible explanation why prevalence was higher in sweat bees than other wild bees (e.g. *Bombus impatiens* and *B. griseocollis*), wherein sweat bees have short tongues (2–6 mm in length; [62]) and bumble bees have long tongues (4–16 mm in length; [61]). A foraging bee's handling time on each flower is positively correlated with the depth of flower heads and the corresponding length of bee tongues required to reach flower nectaries [32, 63, 64]. Therefore, the short-tongued sweat bees, may be faster foragers and able to visit more flowers, thereby increasing the likelihood of contact with infected bees and or contaminated flowers [20]. More research is needed to determine

infectivity of DWV and IAPV in sweat bees, which would clarify whether sweat bees act as asymptomatic carriers of viruses capable of infecting others.

## Did landscape types correlate with virus detection?

Our data showed no significant difference between landscape types; however, our sites were either in or near city limits and not completely isolated. Floral availability measures would have allowed for better distinctions to be made between landscape types but were not taken into consideration for this study. In agricultural landscapes, the rate of virus-infected honey bees increased from early-mid- to late-summer samples. Low sample sizes in wild bees caught during late summer may have contributed to observed differences in the frequency of virus-infected honey and wild bees. Managed bees can be placed in any habitat and moved from one location to another, particularly when floral resources are limited. Wild bees, in contrast, depend on the pockets of natural or semi-natural areas scattered around agricultural and urban areas to provide nesting habitats [65]. Therefore, plant-pollinator community richness and pollination to crop plants by wild bees depends on the proximity of crop fields to natural areas [65]. Diminishing natural habitat and greater pesticide exposure in primarily agricultural areas may explain the low number of wild bees found in agricultural sites. In contrast, no honey bees were collected in open space sites. Both honey bees and wild bees were collected from urban sites but more viruses were detected in wild bees compared to honey bees. Urban gardens typically pack high volumes of diverse plants in concentrated areas. The higher density of common foraging sites may increase the number of bee visitations per flower or the likelihood of bee—bee interactions, which could increase the risk of interspecific virus transmission. Planting flowers in larger and less dense areas may reduce bee—bee interactions and lower the chance of coming into contact with infected bees or contaminated flowers.

## Does pollen play a role in interspecific transmission of viruses?

We hypothesized that if honey bee viruses were transmitted indirectly to different species of bees though shared use of contaminated flowers then viruses would be frequently detected in flowers and floral resources. Pollen collected from corbicula of honey bees and from in-hive pollen showed low rates of DWV and BQCV only. The low frequency of detections yielded results not sufficient to assess the persistence and association of viruses with pollen and would require further research.

## Conclusions

Collectively, our results indicate that prevalence of viruses in bees was highly dependent indicating physiological and or behavioral differences among bee species. Viral prevalence was also affected by virus type and season, but not by landscape or year. Low sample sizes of wild bees and low frequency of virus detections made results difficult to interrupt, whereas clear differences were observed in DWV rates across bee species and seasons, while IAPV was predominately detected in sweat bees and in late season collections. Our data provides greater insight into the prevalence of viruses among bee communities, yet more research is needed to elucidate potential changes in bee and land management practices that could mitigate disease transmission and spread among pollinator communities.

## Supporting information

**S1 Fig. Detection of DWV and IAPV viruses on *Apis mellifera, Bombus impatiens, B. griseocollis, Halictus ligatus* sampled from urban site in early–mid summer in 2018.** Primer

pairs species for DWV and IAPV were used separately to amplify RT-PCR products of 194 and 586 bp, respectively. Negative ($H_2O$) and positive controls were included in each run of the RT-PCR.
(DOCX)

**S2 Fig. Viruses detected in pollen samples.** Virus prevalence, or the number of positive detections of BQCV, DWV, IAPV, and or SBV, in pollen loads collected from the legs of foraging honey bees or from in-hive pollen stores collected in early and late summer (2017–2018).
(DOCX)

**S1 Table. Description of the three landscape types (Agricultural, Urban, and Natural/ Open) and location information for collection sites from 2017 and 2018.** Sites with "*" denotes larger locations that had multiple distinct collection areas; however, sampling effort was standardized across landscape types and across season.
(DOCX)

## Acknowledgments

We thank Kelsey Karnik (University of Nebraska-Lincoln) and Prof. Çiğdem Takma (Ege University) for providing statistical support and Kayla Mollet and Katie Lamke for assisting in field collections and sample processing.

## Author Contributions

**Conceptualization:** Tugce Olgun, Judy Wu-Smart.

**Data curation:** Tugce Olgun, Judy Wu-Smart.

**Formal analysis:** Tugce Olgun, Sydney E. Everhart, Judy Wu-Smart.

**Funding acquisition:** Judy Wu-Smart.

**Investigation:** Tugce Olgun.

**Methodology:** Tugce Olgun, Troy Anderson, Judy Wu-Smart.

**Project administration:** Tugce Olgun, Judy Wu-Smart.

**Resources:** Tugce Olgun, Troy Anderson, Judy Wu-Smart.

**Software:** Tugce Olgun, Sydney E. Everhart.

**Supervision:** Judy Wu-Smart.

**Validation:** Tugce Olgun, Sydney E. Everhart, Judy Wu-Smart.

**Visualization:** Tugce Olgun, Judy Wu-Smart.

**Writing – original draft:** Tugce Olgun.

**Writing – review & editing:** Sydney E. Everhart, Troy Anderson, Judy Wu-Smart.

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
