## [Decision Letter · Decision Letter 0]

5 Feb 2020

PONE-D-19-35401

Comparative analysis of viruses in four bee species collected from agricultural, urban, and natural landscapes

PLOS ONE

Dear Dr. Wu-Smart,

Thank you for submitting your manuscript to PLOS ONE. After careful consideration, we feel that it has merit but does not fully meet PLOS ONE’s publication criteria as it currently stands. Therefore, we invite you to submit a revised version of the manuscript that addresses the points raised during the review process (for details, see below). Please pay special attention to the major points raised by Reviewer #2.

We would appreciate receiving your revised manuscript by March 15th, 2020. To enhance the reproducibility of your results, we recommend that if applicable you deposit your laboratory protocols in protocols.io, where a protocol can be assigned its own identifier (DOI) such that it can be cited independently in the future. For instructions see: http://journals.plos.org/plosone/s/submission-guidelines#loc-laboratory-protocols

We look forward to receiving your revised manuscript.

Kind regards,

Wolfgang Blenau

Academic Editor

PLOS ONE

"This project is based on research supported by The Republic of Turkey Ministry of National Education and partially supported by the Nebraska Department of Agriculture Specialty Crop Block Grant funding program (Accession Number 18-13-314) from the USDA Agricultural Marketing Service and by the Nebraska Agricultural Experiment Station with funding from the Hatch Multistate Research capacity funding program (Accession Number 1011128) from the USDA National Institute of Food and Agriculture."

Reviewers' comments:

Reviewer's Responses to Questions

**Comments to the Author**

1. Is the manuscript technically sound, and do the data support the conclusions?

Reviewer #1: Yes

Reviewer #2: Partly

2. Has the statistical analysis been performed appropriately and rigorously? 

Reviewer #1: Yes

Reviewer #2: No

3. Have the authors made all data underlying the findings in their manuscript fully available?

Reviewer #1: Yes

Reviewer #2: No

4. Is the manuscript presented in an intelligible fashion and written in standard English?

Reviewer #1: Yes

Reviewer #2: Yes

5. Review Comments to the Author

Reviewer #1: Editors,

The manuscript entitled, “Comparative analysis of viruses in four bee species collected from agricultural, urban, and natural landscapes” by Olgun et. al describes virus prevalence in managed and unmanaged bee species in multiple landscapes during two ranges of sampling dates (i.e., May 1 – July 30, and Aug. 1 – Sept. 1).

The introduction is informative and well written. The study is important to understanding the ecology of bee viruses in different landscapes. In the long term, studies like this will contribute to the development of management strategies to reduce losses of both managed and wild bee species attributed to virus infection.

Minor points to clarify or address before publication include:

1. Table 1 – It would be good to include sample size for each species and sampling season.

Maybe putting a “(n=x)” below the percentage in each cell. If all sample sizes are uniform, it can just be stated in the caption (in addition to text and methods). This information for honey bees is included in Table 5.

2. Tables 1 and 2 – present data from the same sample cohort – correct?

If not, it would be good to break down the “managed” and “unmanaged” bees by species – but I think Table 2 does that.

3. Lines 360 – 365. The description could include statements about which group of bees had a higher virus prevalence. Table 2 presents the virus prevalence data and Table 3 presents the statistics – but it was difficult to look at both tables at once, and then just read in the text that they were different.

4. Lines 373 – 378 Likewise, the description of the data in the text could include statements of the significant results (i.e., state how they comparisons differed, rather than only that they differed).

5. Table 7, the caption indicates that this table presented the mean number of viruses in all bee species present. I think this means all the positive tests (for 4 viruses) over the total number of PCR tests carried out – correct?

6. Lines 406 – 407 describe Table 7. I think the text is described as percentages and the table as decimal values – it may be good to use the same for both.

7. Table 6 and associated text (477-479 – It isn’t clear to me how Table 6 shows that the number of mono/di/trii infections was higher in agricultural sites. I think the majority of honey bee colonies (or samples from honey bee colonies) were obtained from colonies located in agricultural sites – correct?

8. Table 9 – fonts may differ from the text and figures (of course changing this is optional).

9. Suggest citing:

(1) Glenny et al PONE 2017 - doi.org/10.1371/journal.pone.0182814

Since this study also determined the sample date is most correlated to virus prevalence and abundance in managed honey bee colonies.

(2) Grozinger and Flenniken, Ann Rev Entomology 2019

doi.org/10.1146/annurev-ento-011118-111942

Bee virus ecology, even substituted for Brutscher et al PLOS Pathogens.

10. Line 529 – change the word “spill over” – since it isn’t clear that the viruses “spill over” from honey bees to other bees.

Viruses that infect honey bees, are called “honey bee viruses” since that is the species in which they were discovered. Honey bees are also the most studied species – in terms of virus prevalence and abundance. While transmission direction can be surmised by greater prevalence in one species of sympatric species – it would be best shown from temporal data from the same site, since virus infection in bees are quite dynamic (see Glenny et al 2017).

Also, while some studies (e.g., Furst, Paxton, Brown, et al. Nature 2014 ) from the UK suggest DWV transmission from honey bees to bumble bees, additional data (maybe obtained from the same sample cohort) indicate that another RNA virus (ABPV) was more prevalent in wild bumble bees than in sympatric honey bee samples.

As stated in the introduction of this manuscript, viruses are transmitted between different genera / species of bees – so that use of the word “spill over” is not appropriate. Though, there may be several definitions of the word “spillover” (e.g., Wikipedia “Spillover infection, also known as pathogen spillover and spillover event, occurs when a reservoir population with a high pathogen prevalence comes into contact with a novel host population. The pathogen is transmitted from the reservoir population and may or may not be transmitted within the host population.)” doesn’t seem accurate for bee viruses. Additional temporal studies are required to determine the ecology of bee viruses. Particularly, since in Lines 554-556 of this study wild bees had higher prevalence of virus than honey bees in urban settings.

Line 564-566 could be revised given the comments above.

Furst MA, McMahon DP, Osborne JL, Paxton RJ, Brown MJF. Disease associations between honeybees and bumblebees as a threat to wild pollinators. Nature. 2014;506(7488):364–+. pmid:WOS:000331477800040. “ inferred DWV from honey bees to bumble bees”

McMahon DP, Furst MA, Caspar J, Theodorou P, Brown MJF, Paxton RJ. A sting in the spit: widespread cross-infection of multiple RNA viruses across wild and managed bees. J Anim Ecol. 2015;84(3):615–24. pmid:WOS:000353405300004.

“inferred ABPV from wild bees to honey bees”

11. Line 518 - delete extra “.” before citations

Reviewer #2: This study examines prevalence of 4 common bee viruses in two (potentially) managed bees and two wild species in a variety of environments in Nebraska across 3 time points. This study adds to the body of work on bee pathogens in the environment, but in the current version, there are major issues with data analysis that prevent reviewing the results and discussion.

Major points

- The manuscript seems to frequently confound prevalence with transmission; without sequence data, it is really unclear whether particular prevalence patterns indicate transmission between species, although they can give some rough pointers to transmission potential

- Finding DWV and IAPV mainly in Halictus is really interesting, but it is not clear from the ms whether these are true infections. For this part – reporting that these viruses are predominantly found in Halictus – it would really be necessary to run at least a proof-of-principle test using strain-specific rtPCR (and arguably also for the two bombus species)

- It is unclear whether the collected B. impatiens bees were from commercial or from wild colonies; is there any evidence to support that these are predominantly from managed colonies?

- The information on field sites needs to be clarified, it is rather unclear how many sites were used (3 in agricultural land? 4 in natural areas?); a table with sites and their characteristics, and very importantly how distant they are from each other would help; this is also missing in the results section

- Honeybee colonies were moved to several of the sampling sites; were they tested ahead of being moved? Could they have changed the prevalence (and strain) of viruses circulating locally?

- Most importantly, the current analysis of prevalence data is not suitable, you need to do GLMs to test for differences in prevalence to incorporate the different explanatory variables (including if relevant managed/unmanaged) rather than doing individual chi-squared tests!

Given the analytical issues that need to be resolved (no GLMs; additionally reason for distinction between managed/unmanaged unclear), I have not looked at the results and discussion section in detail.

Methods issues:

- Bee dissection: this makes it sound like as though you only dissected 10 bees in total per species, that obviously can’t be right. But does it mean that you did not test all of the sampled bees? Sample size is very unclear!

- It’s really not ideal to store samples for RNA virus studies at -20

- Pollen collection: again, sample size is unclear. Pollen traps were used on trhee hives at UNL (presumably both years). But is unclear where and how many samples from flowers and bees were collected; does the collection from hair refer to sweat bees only? How sensitive is the analysis when using a bit of pollen from a flower or pollen collected from hairs (as opposed to quite massive pollen pellets from corbiculate bees)?

- Limiting samples to the thorax may also reduce detection for viruses prevalent in the nervous system (DWV!) and for fecally or orally transmitted viruses

- I am surprised that only a ‘barely visible pellet’ of RNA was extracted using this protocol

- Were the primers specific to DWV-A or B? Would they pick up DWV-B infections (on the rise in the US according to a recent paper by Ryabov et al)

Analysis

- Prevalence of viruses across landscapes needs to be analysed by GLMs, the present analysis is not suitable!

Abstract

Lines 32 -35: I don’t understand why the pattern described for DWV should indicate higher interspecific transmission – particularly as it’s not clear what that comparison is too (IAPV – would not make any sense); the seasonal difference of 10.8 to 11.4% is almost certainly negliglible

Lines 35-37: I assume that this is explained more fully in the ms, but in conjunction with the sentence above, it doesn’t make much sense to put the numbers in here; also, I was wondering what’s up with BQCV and SBV?

Intro

- First paragraph: I really enjoyed reading this section, but given that you also looked at B. griseacolis and Halictus ligatus, it should really also include a section on these species (or on the wider importance of wild bees)

- Line 119: ‘in only urban site’ – unclear, is this meant to say ‘only in urban sites’?

Line 197: the role pollen plays

Line 207: processed and assessed

Line 575: interp

6. PLOS authors have the option to publish the peer review history of their article (what does this mean?). If published, this will include your full peer review and any attached files.

Reviewer #1: No

Reviewer #2: No

---

## [Author Response · Author response to Decision Letter 0]

14 May 2020

Deposit the laboratory protocols:

DOI Number: dx.doi.org/10.17504/protocols.io.bdh6i39e

Editor :

Thank you for stating the following in the Acknowledgments Section of your manuscript:

"This project is based on research supported by The Republic of Turkey Ministry of National Education and partially supported by the Nebraska Department of Agriculture Specialty Crop Block Grant funding program (Accession Number 18-13-314) from the USDA Agricultural Marketing Service and by the Nebraska Agricultural Experiment Station with funding from the Hatch Multistate Research capacity funding program (Accession Number 1011128) from the USDA National Institute of Food and Agriculture." We note that you have provided funding information that is not currently declared in your Funding Statement. However, funding information should not appear in the Acknowledgments section or other areas of your manuscript. We will only publish funding information present in the Funding Statement section of the online submission form. Please remove any funding-related text from the manuscript and let us know how you would like to update your Funding Statement. Currently, your Funding Statement reads as follows:

Response: We have addressed this and changed the Acknowledement Section.

Reviewer 1: 

1. Table 1 – It would be good to include sample size for each species and sampling season.

Maybe putting a “(n=x)” below the percentage in each cell. If all sample sizes are uniform, it can just be stated in the caption (in addition to text and methods). This information for honey bees is included in Table 5.

Response: We have included (n/total bees) to Table 1 to indicate sample size for each species.

2. Tables 1 and 2 – present data from the same sample cohort – correct?

If not, it would be good to break down the “managed” and “unmanaged” bees by species – but I think Table 2 does that.

Response: The reviewer is correct in that Table 1 and 2 represent the same cohort. And given one of our main research question is regarding the comparison among managed vs unmanaged species, we feel Table 2 is necessary to portray that information. No additional changes were suggested by the reviewer.

3. Lines 360 – 365. The description could include statements about which group of bees had a higher virus prevalence. Table 2 presents the virus prevalence data and Table 3 presents the statistics – but it was difficult to look at both tables at once, and then just read in the text that they were different.

Response: We have added visual indicators to Table 2 to denote significance among pairings that combines the information from Table 3 into Table 2 for easilier interpretation of results as suggested by the reveiwer.

4. Lines 373 – 378 Likewise, the description of the data in the text could include statements of the significant results (i.e., state how they comparisons differed, rather than only that they differed).

Response: We have stated how the comparisons differed in lines 361-372.

5. Table 7, the caption indicates that this table presented the mean number of viruses in all bee species present. I think this means all the positive tests (for 4 viruses) over the total number of PCR tests carried out – correct?

Response: Table 7 shows the mean number of viruses detected in all bee species. We have added “positive” in the caption of table to emphasise infected bees with viruses more clearly as suggested. “Table 7. The mean number (± s.d.) of positive viruses present in sampled bee species by landscape types, season, and year.”

6. Lines 406 – 407 describe Table 7. I think the text is described as percentages and the table as decimal values – it may be good to use the same for both.

Response: We have changed the percentages with decimal values in Table 7 as suggested.

7. Table 6 and associated text (477-479 – It isn’t clear to me how Table 6 shows that the number of mono/di/trii infections was higher in agricultural sites. I think the majority of honey bee colonies (or samples from honey bee colonies) were obtained from colonies located in agricultural sites – correct?

Response: In 477-479 lines, we provide a comparison between agriculture and urban sites. There were 440 honey bees randomly sampled from 44 colonies in urban site and 150 honey bees sampled from 15 colonies located in agriculture sites. This information has been added into the text (lines 499-504).

8. Table 9 – fonts may differ from the text and figures (of course changing this is optional). 

Response: We have changed fonts in all figures to make fonts consistent.

9.Suggest citing:

(1) Glenny et al PONE 2017 - doi.org/10.1371/journal.pone.0182814

Since this study also determined the sample date is most correlated to virus prevalence and abundance in managed honey bee colonies.

(2) Grozinger and Flenniken, Ann Rev Entomology 2019

doi.org/10.1146/annurev-ento-011118-111942

Bee virus ecology, even substituted for Brutscher et al PLOS Pathogens

Response: We have included Glenny et al. 2017 in discussion section.

10. Line 529 – change the word “spill over” – since it isn’t clear that the viruses “spill over” from honey bees to other bees.

Response: Taking reviewer’s suggestions, we have changed the word “spill over”” with the word “prevalence”.

Reviewer 2:

Finding DWV and IAPV mainly in Halictus is really interesting, but it is not clear from the ms whether these are true infections. For this part – reporting that these viruses are predominantly found in Halictus – it would really be necessary to run at least a proof-of-principle test using strain-specific rtPCR (and arguably also for the two bombus species)

Response: To report that these viruses are predominantly found in bee species, we have run a proof-of-principle test using strain-specific rtPCR and have included the image as Supplementary Figure 1 (S1 Fig.). 

S1 Fig. Detection of DWV and IAPV viruses on Apis mellifera, Bombus impatiens, B. griseocollis, Halictus ligatus sampled from urban site in early –mid summer in 2018. Primer pairs species for DWV and IAPV were used separately to amplify RT-PCR products of 194 and 586 bp, respectively. Negative (H2O) and positive controls were included in each run of the RT-PCR.

It is unclear whether the collected B. impatiens bees were from commercial or from wild colonies; is there any evidence to support that these are predominantly from managed colonies?

Response: We collected both wild and commercial B. impatiens at field sites. There were 4-6 commerical B. impatiens colonies along the open roadside sites, however, to distingiush whether field caught bumble bees were from commerical or wild sources was beyond the scope of this study. We have included more information to clarify in lines 168-172.

The information on field sites needs to be clarified, it is rather unclear how many sites were used (3 in agricultural land? 4 in natural areas?); a table with sites and their characteristics, and very importantly how distant they are from each other would help; this is also missing in the results section

Response: To clarify field sites, we included a supplementary table to provide additional information (S1 Table). We have also included more information under “target bee species and site selection”. 

S1 Table. Description of the three landscape types (Agricultural, Urban, and Natural/Open) and location information for collection sites from 2017 and 2018. Sites with “*” denotes larger locations that had multiple distinct collection areas, however, sampling effort was standized across landscape types and across season. 

Honey bee colonies were moved to several of the sampling sites; were they tested ahead of being moved? Could they have changed the prevalence (and strain) of viruses circulating locally?

Response: In line 187-197, we provided information about how many hives were moved from pollinator garden to other agricultural sites. We did not check the viral prevalence in wild bees at those sites before the honey bee hives were moved. And while it is possible that prevalence (and strain) of viruses would be circulated locally, these apiaries are used each year so viruses could be circulating from previous years. Further, there were no detection of viruses at these sites until late summer, months after the hives were moved to sites, indicating this was not the case. 

Most importantly, the current analysis of prevalence data is not suitable, you need to do GLMs to test for differences in prevalence to incorporate the different explanatory variables (including if relevant managed/unmanaged) rather than doing individual chi-squared tests!

Response: We worked with two different statisticians in this study. Both of them suggested to use chi-squared tests because GLMs analysis was not suitable for the zero-inflated data set. We have reanalyzed the data using a negative binomial logistic regression model to verify our orginal appoarch and have included this output. Results from this regression analysis provided similiar results to our orginal chi square analyses, however, is unable to compared data among managed and unmanaged treatment groups therefore we feel confident the appoarch we used was suitable and appropriate for this study. 

Methods

(Bee dissection): this makes it sound like as though you only dissected 10 bees in total per species, that obviously can’t be right. But does it mean that you did not test all of the sampled bees? Sample size is very unclear!

Response: We have added some details about bee tissue dissection on line 204-213 to make samples size clear. 

It’s really not ideal to store samples for RNA virus studies at -20

Response: While in retrospect we agree that it is not ideal to store samples at -20 for RNA virus studies, there are many studies that reference the methods we used while processing samples (-20 for short term storage) and then -80 for long term storage. Some of these references include: 

Glenny W, Cavigli I, Daughenbaugh KF, Radford R, Kegley SE, Flenniken ML (2017) Honey bee (Apis mellifera) colony health and pathogen composition in migratory beekeeping operations involved in California almond pollination. PLoS ONE 12(8): e0182814. https://doi.org/10.1371/journal.pone.0182814

Fürst, M., McMahon, D., Osborne, J. et al. Disease associations between honeybees and bumblebees as a threat to wild pollinators. Nature 506, 364–366 (2014). https://doi.org/10.1038/nature12977

Dolezal, A. G., Hendrix, S. D., Scavo, N. A., Carrillo-Tripp, J., Harris, M. A., Wheelock, M. J., ... & Toth, A. L. (2016). Honey bee viruses in wild bees: viral prevalence, loads, and experimental inoculation. PloS one, 11(11).

Pollen collection: again, sample size is unclear. Pollen traps were used on trhee hives at UNL (presumably both years). But is unclear where and how many samples from flowers and bees were collected; does the collection from hair refer to sweat bees only? How sensitive is the analysis when using a bit of pollen from a flower or pollen collected from hairs (as opposed to quite massive pollen pellets from corbiculate bees)?

Response: We have decided to not include pollen collected from flower and hairs because of limited sample size. Therefore, we adjusted (line 220-223) the narrative and provided additional supplementary figure (S2 Fig.) to clarify. 

S2 Fig Viruses detected in pollen samples. Viral prevalence, or the number of positive detections of BQCV, DWV, IAPV, and or SBV, in pollen loads collected from the legs of foraging honey bees or from in-hive pollen stores collected in early and late summer (2017–2018). 

Limiting samples to the thorax may also reduce detection for viruses prevalent in the nervous system (DWV!) and for fecally or orally transmitted viruses

Response: It is possible to detect DWV in the head but as Boncristiani et al. (2011), to reduce the likelihood of a false negative virus detection in PCR due to known inhibitory substances present in the compound eyes and guts of honey bees and insects, thorax of bees was used as described under bee tissue dissection in the manuscript. 

Boncristiani, H, Li J, Evans J, Pettis J, Chen Y. Scientific note on PCR inhibitors in the compound eyes of honey bees, Apis mellifera. Apidologie. 2011;42: 457-460. doi: 10.1007/s13592-011-0009-9

I am surprised that only a ‘barely visible pellet’ of RNA was extracted using this protocol

Response: We extracted RNA using the described protocol succesfully. The term has been adjust to “very small”.

Were the primers specific to DWV-A or B? Would they pick up DWV-B infections (on the rise in the US according to a recent paper by Ryabov et al)

Response: Recently, studies indicate multiple strains or variants of Deformed wing virus (DWV) exists which include type A (DWV; Lanzi et al. 2006 and Kakugo virus; Fujiyuki et al. 2004), type B (Varroa destructor virus-1 (VDV-1); Martin et al. 2012, Mordecai et al. 2015), and type C (phylogenetically distinct from type A and B; Mondecai et al. 2016)) (Mondecai et al. 2016). In this study, the primers were not specifically DWV-A or B. We modeled methodology after the Chen et al. and there were no analyses of DWV variants in this study. We have added some informations about DWV variants and clarify that in this study, DWV variants were not assessed.

Lines 32 -35: I don’t understand why the pattern described for DWV should indicate higher interspecific transmission – particularly as it’s not clear what that comparison is too (IAPV – would not make any sense); the seasonal difference of 10.8 to 11.4% is almost certainly negliglible

Response: “The higher prevalence of DWV detected across bee species (10.4 % on Apis mellifera, 5.3 % on Bombus impatiens, 6.1 % on Bombus griseocollis, and 22.44 % on Halictus ligatus) and seasons (10.8 % in early-mid summer and 11.4 % in late summer) indicated a higher risk of interspecific transmission of DWV.”

In lines 33-36, we provide a comparison to show all 4 bee species had greater detection rates of DWV in both early-mid summer and late summer indicating that DWV was more readily found in all sampled bee types in the same area which suggests interspecific transmission of DWV. Table1 shows that IAPV as well as DWV was observed on mostly Halictus ligatus and Apis mellifera, but there is limited detection of other virus types on 4 different bee species. That is why only DWV and IAPV were predominately described in the abstract. Further, the Kruskal-Wallis test (using a Bray-Curtis dissimilarity matrix and Dunn’s test) show that DWV and IAPV were significantly different between Halictus ligatus and all other bee species (A. mellifera, B. impatiens, and B. griseocollis). SBV and BQCV are not significant for virus species. No changes suggested so we modified some text to the abstract to clarify (lines 38-40).

Lines 35-37: I assume that this is explained more fully in the ms, but in conjunction with the sentence above, it doesn’t make much sense to put the numbers in here; also, I was wondering what’s up with BQCV and SBV?

Response: We have added “However, there were limited detections of SBV and BQCV in bees collected during both sampling periods, indicating SBV and BQVV may be less prevalent among bee communities in this area.” on abstract on lines 38-40.

Intro

First paragraph: I really enjoyed reading this section, but given that you also looked at B. griseacolis and Halictus ligatus, it should really also include a section on these species (or on the wider importance of wild bees)

Response: Reviewer is right and we have included more information about B. griseocollis and Halictus ligatus in line 61-67.

Line 119: ‘in only urban site’ – unclear, is this meant to say ‘only in urban sites’?

Response: We have corrected ‘in only urban site’ as ‘only in urban sites’. 

Line 197: the role pollen plays

Response: We have corrected “the role pollen plays”.

Line 207: processed and assessed

Response: We have corrected “processed and assessed”.

Line 575: interp

Response: We could not find this.

---

## [Decision Letter · Decision Letter 1]

27 May 2020

Comparative analysis of viruses in four bee species collected from agricultural, urban, and natural landscapes

PONE-D-19-35401R1

Dear Dr. Wu-Smart,

We are pleased to inform you that your manuscript has been judged scientifically suitable for publication and will be formally accepted for publication once it complies with all outstanding technical requirements.

With kind regards,

Wolfgang Blenau

Academic Editor

PLOS ONE

Additional Editor Comments (optional):

Reviewers' comments:

Reviewer's Responses to Questions

**Comments to the Author**

1. If the authors have adequately addressed your comments raised in a previous round of review and you feel that this manuscript is now acceptable for publication, you may indicate that here to bypass the “Comments to the Author” section, enter your conflict of interest statement in the “Confidential to Editor” section, and submit your "Accept" recommendation.

Reviewer #1: All comments have been addressed

2. Is the manuscript technically sound, and do the data support the conclusions?

Reviewer #1: Yes

3. Has the statistical analysis been performed appropriately and rigorously? 

Reviewer #1: Yes

4. Have the authors made all data underlying the findings in their manuscript fully available?

Reviewer #1: Yes

5. Is the manuscript presented in an intelligible fashion and written in standard English?

Reviewer #1: Yes

6. Review Comments to the Author

Reviewer #1: The authors addressed all of my comments. I wonder if the suggestion for "strain-specific" PCR was actually for "strand-specific" PCR to document virus replication (infection) vs. detection.

Strand-specific PCR would be a great addition to the manuscript, but it is not required.

7. PLOS authors have the option to publish the peer review history of their article (what does this mean?). If published, this will include your full peer review and any attached files.

Reviewer #1: No

---

## [Editor Report · Acceptance letter]

4 Jun 2020

PONE-D-19-35401R1 

Comparative analysis of viruses in four bee species collected from agricultural, urban, and natural landscapes 

Dear Dr. Wu-Smart:

I'm pleased to inform you that your manuscript has been deemed suitable for publication in PLOS ONE. Congratulations! Your manuscript is now with our production department. 

Kind regards, 

on behalf of

Dr. Wolfgang Blenau 

Academic Editor

PLOS ONE